# RB depletion is required for the continuous growth of tumors initiated by loss of RB

**Alex Doan**[1,2☯], **Julia Arand**[1,2,3☯], **Diana Gong**[1,2,4], **Alexandros P. Drainas**[1,2], **Yan Ting Shue**[1,2], **Myung Chang Lee**[1,2], **Shuyuan Zhang**[5], **David M. Walter**[6,7], **Andrea C. Chaikovsky**[1,2], **David M. Feldser**[6,7], **Hannes Vogel**[8], **Lukas E. Dow**[9,10,11], **Jan M. Skotheim**[5], **Julien Sage**[1,2]*

**1** Department of Pediatrics, Stanford University, Stanford, California, United States of America, **2** Department of Genetics, Stanford University, Stanford, California, United States of America, **3** Center of Anatomy and Cell Biology, Department of Cell and Developmental Biology, Medical University of Vienna, Vienna, Austria, **4** Department of Bioengineering, Stanford University, Stanford, California, United States of America, **5** Department of Biology, Stanford University, Stanford, California, United States of America, **6** Department of Cancer Biology, Perelman School of Medicine, University of Pennsylvania, Philadelphia, Pennsylvania, United States of America, **7** Abramson Family Cancer Research Institute, Perelman School of Medicine, University of Pennsylvania, Philadelphia, Pennsylvania, United States of America, **8** Department of Pathology, Stanford University, Stanford, California, United States of America, **9** Sandra and Edward Meyer Cancer Center, Weill Cornell Medicine, New York, New York, United States of America, **10** Department of Biochemistry, Weill Cornell Medicine, New York, New York, United States of America, **11** Department of Medicine, Weill Cornell Medicine, New York, New York, United States of America

☯ These authors contributed equally to this work.
* julsage@stanford.edu

**Data Availability Statement:** The Gene Expression Omnibus accession number for the RNA-seq data reported in this paper is GSE173749.

## Abstract

The retinoblastoma (RB) tumor suppressor is functionally inactivated in a wide range of human tumors where this inactivation promotes tumorigenesis in part by allowing uncontrolled proliferation. RB has been extensively studied, but its mechanisms of action in normal and cancer cells remain only partly understood. Here, we describe a new mouse model to investigate the consequences of RB depletion and its re-activation *in vivo*. In these mice, induction of shRNA molecules targeting RB for knock-down results in the development of phenotypes similar to *Rb* knock-out mice, including the development of pituitary and thyroid tumors. Re-expression of RB leads to cell cycle arrest in cancer cells and repression of transcriptional programs driven by E2F activity. Thus, continuous RB loss is required for the maintenance of tumor phenotypes initiated by loss of RB, and this new mouse model will provide a new platform to investigate RB function *in vivo*.

## Author summary

The retinoblastoma protein (RB) is a central regulator of the cell cycle. Functional inactivation of RB leads to unchecked proliferation and is a frequent occurrence in human cancer. Experiments with pre-clinical models have clearly demonstrated that loss of RB is a strong promoter of cancer development. However, the mechanisms through which RB carries out its tumor suppressor function remain only partly understood. Here, we describe a new genetically engineered mouse model in which RB expression can be turned

**Funding:** Research reported in this publication was supported by the NIH (CA231997 and CA228413 to J.S.), the Tobacco-Related Disease Research Program (TRDRP 28IR-0046 to J.S. and T30FT0824 to A.P.D.), a Deutsche Forschungsgemeinschaft post-doctoral fellowship (J.A.), a Bass fellowship from the Stanford Child Health Research Institute (J.A.), and the Stanford Bio-X Undergraduate Summer Research Program (D.G). The funders had no role in study design, data collection and analysis, decision to publish, or preparation of the manuscript.

**Competing interests:** I have read the journal's policy and the authors of this manuscript have the following competing interests: J.S. receives research funding from Pfizer. The authors declare no other competing interests.

on or off. Using this controllable RB knock-down system, we investigated both the consequences of RB loss in normal cells and its re-expression in cancer cells. We found that long-term RB knock-down in mice leads to the development of pituitary and thyroid tumors, recapitulating previously reported phenotypes of *Rb* knockout mice, thus validating this new mouse model. Notably, re-expression of RB in pituitary tumors initiated by loss of RB was sufficient to block tumor growth, indicating that these tumors are still dependent on RB loss for their expansion.

## Introduction

Tumors arise from normal cells upon an accumulation of alterations, including genetic and epigenetic loss-of-function events affecting tumor suppressor genes and gain-of-function events affecting oncogenes. A central issue in the cancer biology field is to identify the mechanisms by which these alterations transform normal cells into cancer cells. A better understanding of these mechanisms can lead to the development of effective anti-cancer strategies (reviewed in [1,2]).

Frequent alterations in many cancer types are found in members of the so-called "RB pathway", and this pathway has been serving as a paradigm in cancer research for several decades. Schematically, in this pathway, the retinoblastoma tumor suppressor protein (RB) controls the activity of the E2F family of transcription factors, which are themselves implicated in the transcription of genes whose products are important for cell cycle progression, especially at the G1/S transition of the cell cycle. During normal cell cycle progression, RB is phosphorylated by cyclin-dependent kinases (CDKs) associated with their cyclin partners, and these phosphorylation events release E2F from the inhibitory activity of RB, thereby allowing cells to replicate their DNA and divide. In cancer cells, RB is found to be frequently inactive, either because of direct genetic and epigenetic inactivation of the *RB* gene (also known as *RB1*) but more frequently by alterations that result in increased activity of the RB kinases Cyclin D-CDK4/6 and Cyclin E-CDK2 (reviewed in [3–6]).

How these Cyclin-CDK kinase complexes promote RB inactivation at the molecular/structural level is still not fully understood (reviewed in [7,8]), but the genetic and biochemical understanding of the RB pathway gained in the past 30 years has guided the development of inhibitors of these kinases (reviewed in [9,10]). The underlying idea is that treatment with these inhibitors will result in RB re-activation as a cell cycle inhibitor, thus inhibiting the proliferation of cancer cells. Importantly, a CDK4/6 inhibitor, palbociclib, was approved 5 years ago to treat estrogen receptor-positive (ER+) breast cancer and recent clinical studies show longer overall survival in women treated with palbociclib-fulvestrant (an anti-estrogen) compared to women treated with placebo-fulvestrant [11,12]. Other CDK4/6 inhibitors have shown similar clinical benefits [13,14] and CDK4/6 inhibitors are now being tested in other human cancer types (clinicaltrials.gov, [15,16]). However, inherent or acquired resistance to CDK4/6 inhibitors frequently occurs and a major goal of a number of current clinical and preclinical studies is to identify changes within the extended RB pathway that can lead to this resistance. While these studies have found cases of loss of RB itself or upregulation of Cyclin-CDK activity [17–20], many cases of resistance lack clear molecular basis [21].

These studies with CDK4/6 inhibitors highlight our partial understanding of the consequences of RB re-activation in cancer cells, especially in relevant *in vivo* contexts. Earlier studies on RB re-activation *in vivo* and in culture often relied on overexpression or truncated RB molecules [22–25]. More recently, re-activation of RB in a lung adenocarcinoma model driven

by oncogenic K-RAS and loss of both RB and p53 was achieved using a new mouse allele in which endogenous RB expression can be genetically turned off and then back on [26]. In this context, re-expression of RB reprogrammed cancer cells towards a less aggressive state but only had a transient effect on inhibition of proliferation [18]. While the transcriptional responses to RB re-expression were not investigated in this model, activation of MAPK signaling and CDK2 were identified as mechanisms that can counter the inhibitory effects of RB re-expression on proliferation [18].

Here, we engineered a new mouse allele in which RB can be knocked-down using RNA interference (RNAi) in an inducible fashion. We found that the phenotypes of RB knock-down in this model closely resemble those of *Rb* knockout mice, thereby validating this new allele. We also show that RB re-expression in pituitary tumors initiated by loss of RB is sufficient to block cell cycle progression, indicating that continuous loss of RB in this context is required for long-term tumor growth.

## Results

### Generation of conditional Rb knock-down mice

To generate mice in which RB levels can be manipulated, we used an inducible shRNA system with a miR-E-based, doxycycline-regulated (Tet-ON), GFP-linked shRNA at the *Col1a1* locus [27,28] (**Fig 1A**). We first tested the knock-down (KD) efficiency of 13 candidate shRNAs in

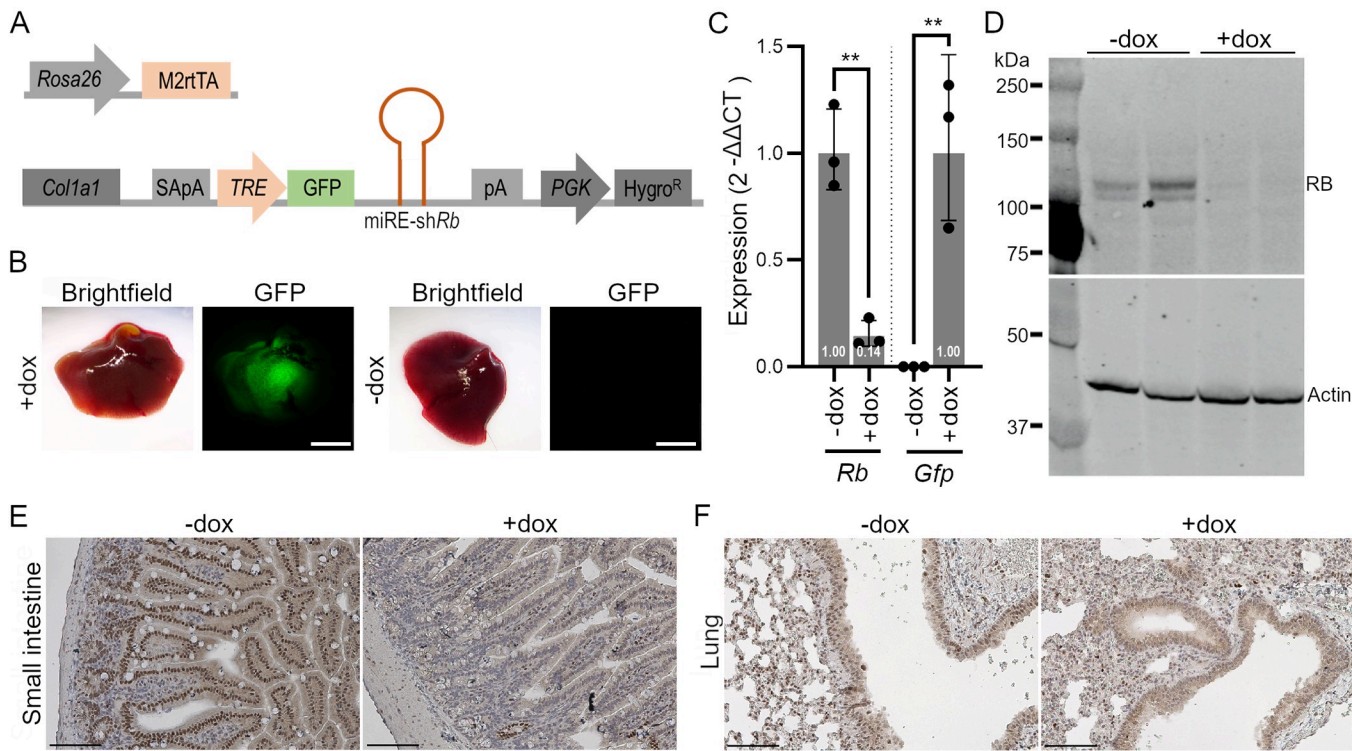

**Fig 1. Generation of sh*Rb* mice to knock-down RB *in vivo*. A.** Schematic representation of the transgenes used to generate sh*Rb* mice. M2-rtTA, M2 reverse tetracycline-controlled transactivator; SA, splice acceptor; pA, polyadenylation signal; TRE, tetracycline response element; GFP, green fluorescence protein; PGK, Phosphoglycerate kinase; Hygro[R], Hygromycin resistance. **B.** Representative images of GFP fluorescent signal in livers of doxycycline (dox)-treated and control sh*Rb* mice 1.5 weeks after dox treatment (n = 2). Scale bar, 4mm. **C.** RT-qPCR analysis of *Rb* and *Gfp* in dox-treated and control livers from sh*Rb* mice 10 days after dox treatment. (n = 3 for each treatment group) (unpaired t test: **, p<0.01). **D:** Immunoblot analysis for RB from liver extracts from dox-treated (n = 2) and control sh*Rb* mice (n = 2), 4 weeks after dox treatment. Actin serves as a loading control. **E-F**: Representative immunohistochemistry images for RB in the small intestine (E) and lungs of 21 days dox-treated (+dox) and control mice (-dox). Note the loss of RB signal in the epithelial cells. Scale bar, 100μm.

mouse NIH3T3 and C2C12 cells and chose sh*Rb*5, which showed the strongest reduction of RB levels (70–80% knock-down) (**S1A** and **S1B Fig** and **S1 Table**). Next, we targeted KH2 mouse embryonic stem cells (mESCs) with the inducible construct. We screened six clones for homogenous GFP expression, RB KD, and normal karyotype, and chose clone sh*Rb*5-4 to generate transgenic mice by blastocyst injections (**S1C** and **S1D Fig**). Chimeric mice were bred and experiments were performed with subsequent generations of *Rosa26^{rtTA} Col1a1^{shRb5-4}* animals, hereafter referred to as sh*Rb* mice. sh*Rb* mice have no obvious developmental phenotypes and are fertile.

We treated adult sh*Rb* mice with doxycycline (dox) in their drinking water to activate rtTA and induce the expression of shRNA molecules targeting *Rb* mRNA molecules (**Fig 1A**). Checking first for GFP expression in different tissues with fluorescence microscopy after 7 days of dox treatment, we detected GFP expression in multiple tissues, as would be expected from the broad pattern of expression of the *Rosa26* promoter driving rtTA expression; no GFP signal was observed in the absence of dox (**Figs 1B** and **S2A**). GFP expression was only weakly detected under the dissecting scope in the lungs or brain of dox-treated mice, respectively, under the same conditions (**S2B Fig**), which was reported before in the same system and is likely due to limited delivery of dox to cells (in the case of brain) or sub-optimal rtTA expression from the *Rosa26* locus (in the case of the lungs) [29,30]. Immunohistochemistry staining showed GFP expression in some cell types in the retina (**S2C Fig**). To determine the efficiency of RB KD *in vivo*, we performed RT-qPCR for both *Rb* and *Gfp* on liver tissue from mice on dox treatment for 10 days. We observed high expression of *Gfp* as well as about 85% knock-down of *Rb* relative to liver samples obtained from non-dox treated controls (**Fig 1C**). Accordingly, RB protein levels were decreased in liver tissue of 28 days dox-treated sh*Rb* mice compared to tissues of untreated control mice (**Fig 1D**). We were not able to reproducibly observe RB by immunohistochemistry staining in the liver (**S2D Fig**). However, the analysis of RB levels by immunohistochemistry in the lung and the small intestine showed consistent knock-down and only a few epithelial cells remained RB-positive after doxycycline treatment (**Fig 1E and 1F**). Together, these experiments validate sh*Rb* mice as a new model to study the effects of controlled RB loss of function in adult mice.

## RB knock-down results in the development of pituitary and thyroid tumors

We next sought to analyze the long-term effects of RB knock-down in sh*Rb* mice. We treated young adult sh*Rb* mice (8–20 weeks of age) with constant dox (+dox) and followed their health status and survival. All sh*Rb* mice (n = 16) died within 18 to 27 weeks after starting dox treatment, while all control mice (missing either one of the two transgenes, +dox, n = 14) survived past 27 weeks (**Figs 2A**, **S3A**, and **3B**). Phenotypic differences between sh*Rb* and control mice initially became noticeable around 10–12 weeks +dox (**Fig 2B** and **2D**, and **S2 Table**). At 12–13 weeks +dox, 4/4 sh*Rb* mice had a scruffy greying hair coat as well as wrinkles of the face, and 2/4 appeared to have an outward curvature of the spine (kyphosis). sh*Rb* mice lost weight with time (**Figs 2C**, **S3C, and S3D**). Overall, these phenotypes were found with increasing severity and frequency in mice at 15–16 and subsequently at 18–19 weeks +dox, in addition to other findings such as alopecia and clouding of the eyes associated with cataract formation (**S4A and S4B Fig**). Control mice treated with dox and sh*Rb* mice without dox treatment showed none of these phenotypes. Thus, dox-associated toxicity and shRNA leakiness are unlikely in this model and the observed phenotypes of dox-treated sh*Rb* mice can be attributed to RB knockdown. Control mice for all subsequent experiments are defined as sh*Rb* mice of similar age without undergoing dox treatment. Upon dissection, we observed a decrease in adipose tissue in the subcutaneous abdominal and peri-gonadal regions as well as profound

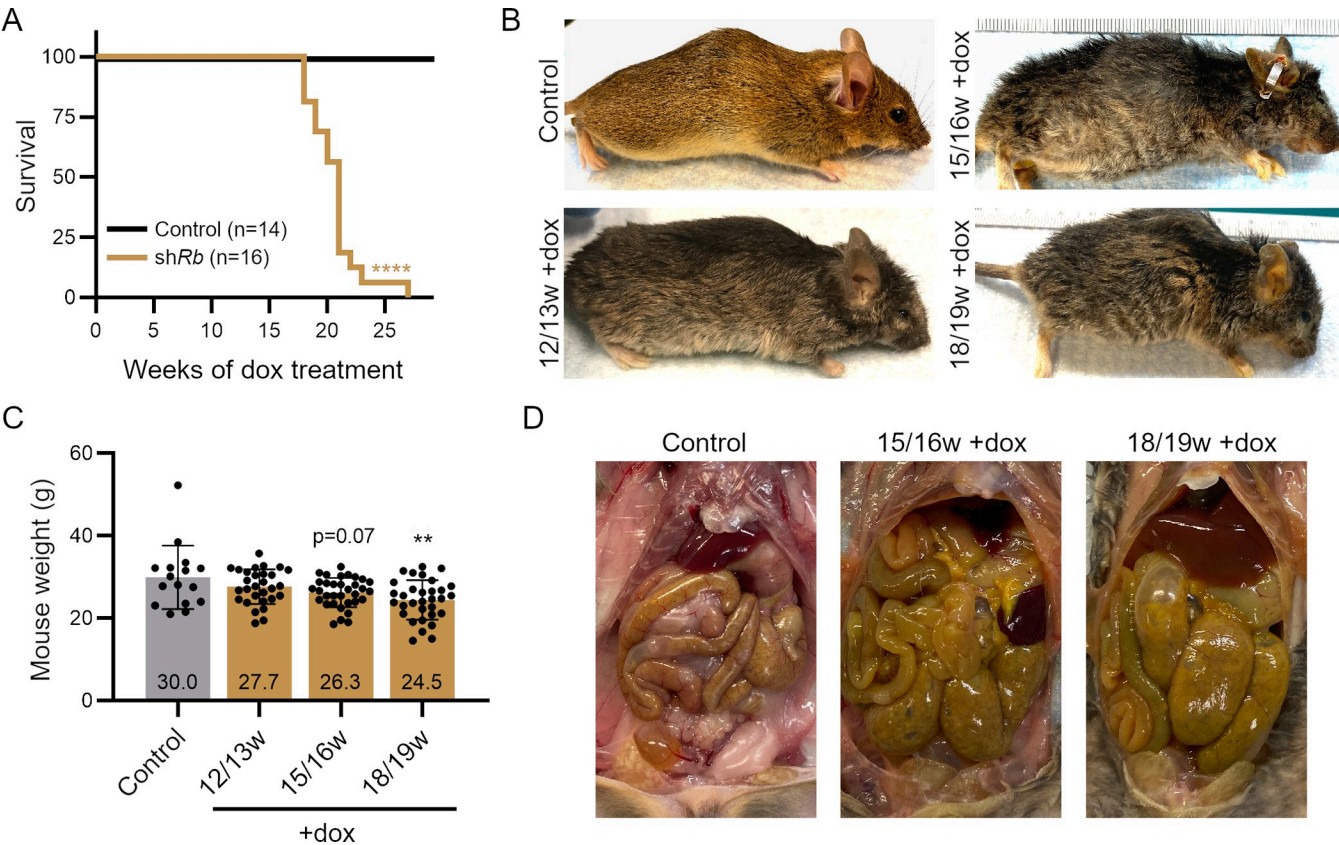

**Fig 2. Phenotypic changes of long-term RB knockdown in sh*Rb* mice. A.** Survival rate of sh*Rb* mice under doxycycline (dox) treatment (n = 14 Control mice, n = 16 sh*Rb* mice) (log-rank (Mantel-Cox) test: ****, p<0.0001). **B.** Representative images of sh*Rb* and control mice after 12–19 weeks (w) of dox treatment. **C.** Weights of sh*Rb* mice after different time points of dox treatment, compared to control mice (age range: 25–35 weeks) (one-way ANOVA analysis: **, p<0.01). **D.** Representative images of internal abdominal cavity in sh*Rb* mice with long term dox treatment. Note organ enlargements (bowels, stomach, and spleen) compared to control mice.

protrusion of the rib cage in sh*Rb* mice from 15–16 weeks +dox onwards (**Fig 2D**). Signs of increased organ size and inflammation were also found throughout the gastrointestinal cavity, with enlargements observed in the stomach, bowels, and spleen (**Fig 2D**). Intestinal sections did not reveal any obvious pathology upon RB knockdown, as described before in knockout mice [31] (**S4C Fig**).

These phenotypic findings induced by the long term continuous systemic loss of RB can have multiple causes, including tumor development. *Rb* KO mice have been reported to develop tumors of the pituitary and thyroid glands [32,33]. sh*Rb* mice showed the occurrence of these same tumors (**Fig 3A**). Pituitary adenomas developed with 100% occurrence in the mice by the time they reached 12–13 weeks of dox treatment, while c-cell thyroid tumors develop relatively slower (25% at 12–13 weeks +dox, 78% at 15–16 weeks +dox) but still reach full occurrence by 18–19 weeks of dox treatment. Weighing of collected pituitary tumors showed drastic mass difference between tumors and normal pituitary glands of control mice, along with a significant continued increase in size of tumors from 15–16 weeks to 18–19 weeks of dox treatment (**Fig 3B**). Histological analysis of harvested pituitary tumors at the various time points showed loss of pituitary gland structure from 12 weeks +dox and the acquisition of tumor morphology phenotypes such as increased cell count and decreased cell size (**Fig 3C**). Immunostaining for the cell cycle marker Ki67 revealed positive proliferation signal at all

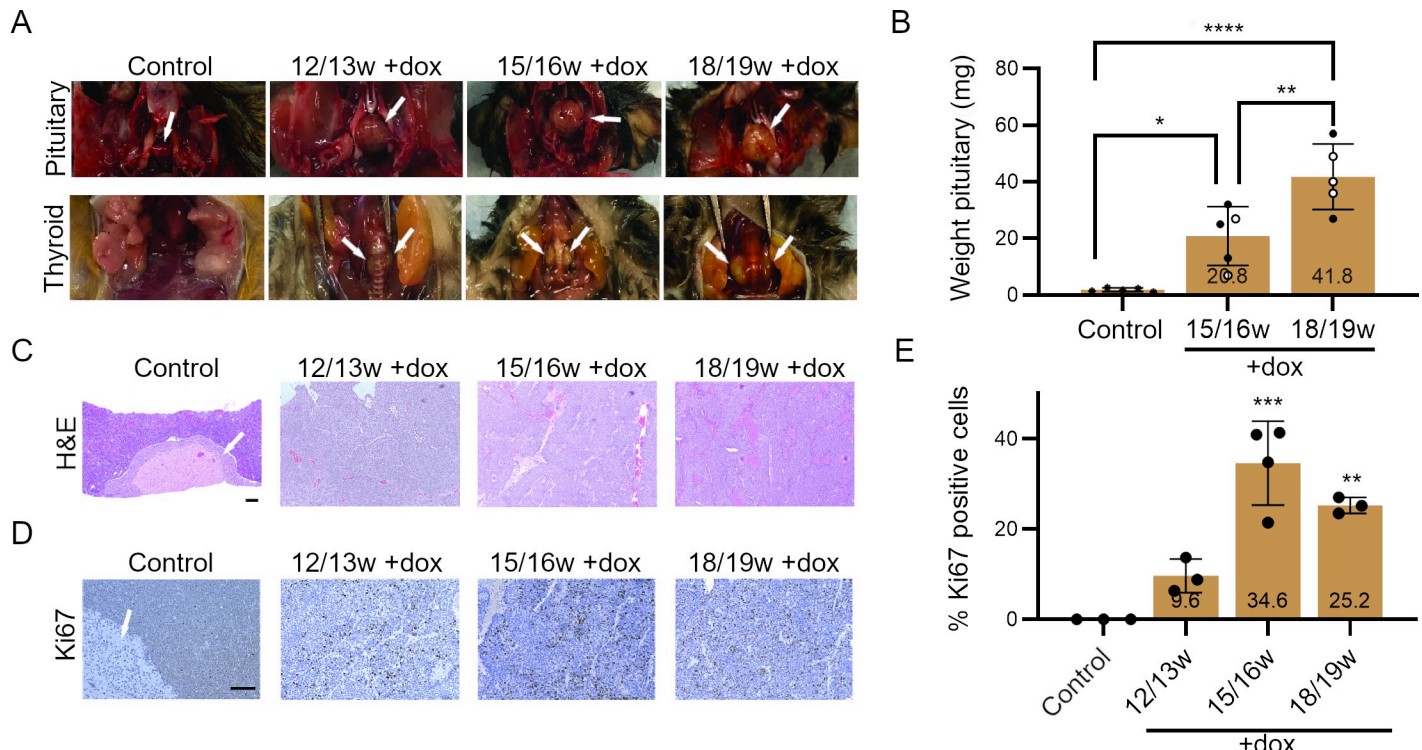

**Fig 3. Development of pituitary and thyroid tumors in shRb mice. A.** Representative images of pituitary and thyroid tumors in sh*Rb* mice at different time points under doxycycline (dox) treatment (w, weeks). Normal pituitary and thyroid glands are no longer distinguishable once the tumors have formed in the skull and surrounding the trachea respectively. **B.** Quantification of pituitary tumor weights at later dox treatment time points compared to control pituitary glands. (Filled circles: males; open circles: females; asterisks: sex of control mice was not determined) (one-way ANOVA analysis: *, p<0.05; **, p<0.01; ****, p<0.0001). **C.** Representative hematoxylin and eosin (H&E) images of sections from pituitary glands/tumors in control mice and in sh*Rb* mice at different time points. The intermediate part of the pituitary gland, from which tumors arise, is shown with an arrow. Scale bar, 100μm. **D.** Representative images of Ki67 immunostaining on sections from pituitary glands in control mice and tumors in sh*Rb* mice at different time points. The intermediate part of the pituitary gland, from which tumors arise, is shown with an arrow. Scale bar, 100μm. **E.** Quantification of Ki67 immunostaining as in (D). Treatment groups were compared to the control group (n = 3 for each treatment group, one way ANOVA analysis: **,p<0.01; ***, p<0.001).

+dox time points within a 5–45% range and highest in samples at the 15/16w and 18/19w time points (**Fig 3D and 3E**). Thyroid tumors were not analyzed in depth but had the expected medullary thyroid cancer histology (**S5 Fig**). Thus, sh*Rb* mice have phenotypes similar to *Rb* knockout mice (summarized in **S3 Table**) and provide an alternative murine model to investigate the development and progression of pituitary and thyroid tumors initiated by loss of *Rb*.

## RB re-expression in RB-deficient cancer cells *in vivo*

One advantage of the inducible KD model over KO models is the possibility to re-express RB through temporal control of dox treatment. We next sought to determine how long it takes *in vivo* upon dox removal to observe re-expression of RB. We first chose to begin rescue experiments in cohorts of mice reaching 16–19 weeks of dox treatment, a window of time where mice show signs of morbidity and pituitary tumors are large. A previous analysis of luciferase KD and re-expression using the same inducible shRNA transgene strategy as in *shRb* mice showed increased expression of luciferase 4 days after dox removal but also that 12 days were needed for more substantial re-expression [29]. When we examined GFP expression in the liver of sh*Rb* mice after dox removal, we found that GFP levels started to decrease after 5 days and kept decreasing until 21 days (**Fig 4A**). These observations suggested that it might take at least 4–5 days to start detecting RB re-expression upon dox removal. To test this idea, we

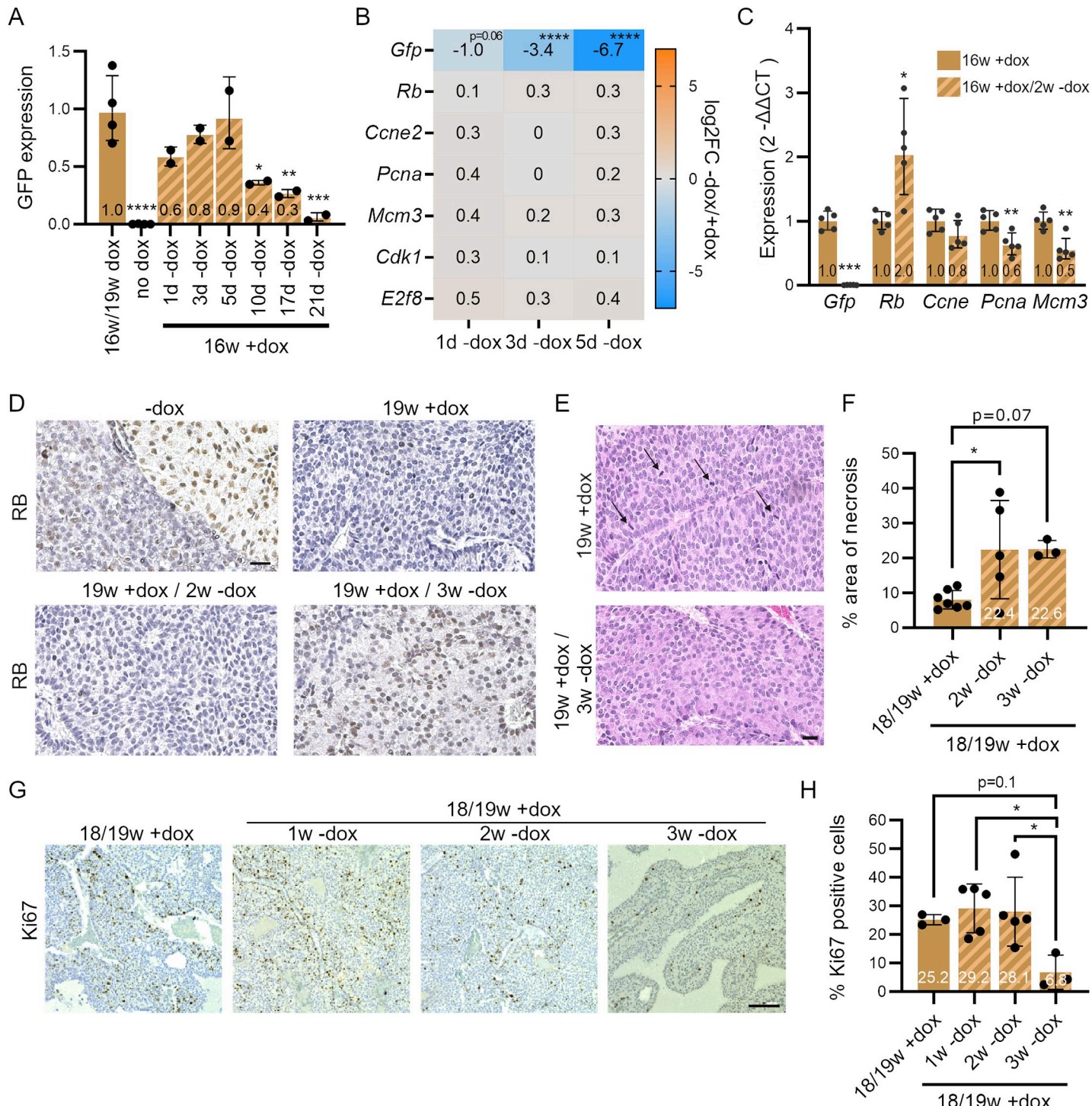

**Fig 4. Kinetics of RB re-expression *in vivo* in sh*Rb* mice. A.** GFP expression measured by quantitative immunoassay on protein extracts from the liver of dox-treated sh*Rb* mice at different time-points (w, weeks) after doxycycline (dox) removal. Signal intensity was normalized to HSP90 (one-way ANOVA, significance is indicated compared to 16w/19w +dox: *, p<0.05; **p<0.01; **, p<0.01; ****, p<0.0001). **B.** Expression changes of *Gfp* and selected cell cycle genes in pituitary RNA from sh*Rb* mice, 1, 3 and 5 days after dox removal compared to dox-treated samples. Expression was determined using RNA-seq (DEseq2: ****, p.adj.<0.0001). **C.** Expression changes of *Gfp* and selected cell cycle genes by RT-qPCR in pituitary RNA from dox-treated sh*Rb* at 16 weeks compared to mice after 2 weeks off dox (two-tailed t-test: *, p<0.05; **p<0.01). **D.** Representative immunostaining images for RB on pituitary section from control mice (-dox), dox-treated sh*Rb* mice (19w +dox), and after dox release (2 or 3 weeks). Scale bar, 20μm. **E.** Representative hematoxylin and eosin (H&E) images of pituitary sections from 19w +dox mice and 19w +dox mice after 3 weeks off dox. Arrows point towards mitotic figures that can be observed in the 19w dox-treated samples. Scale bar, 20μm. **F.** Quantification of necrotic tissue area on section from pituitary tumors from dox-treated sh*Rb* mice or 2 and 3 weeks off dox (one-way ANOVA analysis, significance is indicated compared to 18w/19w +dox: *,

p<0.05). **G.** Representative images of Ki67 immunostaining on sections from pituitary tumors in sh*Rb* mice (18/19w +dox) and 1, 2, and 3 weeks off dox. Scale bar, 100μm. **H.** Quantification of (G) (one-way ANOVA, significances are indicated compared to 18w/19w +dox: *: **,p<0.01; ***, p<0.001).

performed RNA sequencing on pituitary tumors at early 1, 3, and 5 days rescue time points. These data showed progressively decreasing *Gfp* mRNA levels, but *Rb* mRNA levels did not significantly increase at this early stage, and we also observed no changes in the levels of cell cycle genes (**Fig 4B**). We next performed RT-qPCR on tumors at a later rescue time point (14 days) for *Gfp*, *Rb*, and cell cycle genes. At this time point, there was no *Gfp* mRNA detectable anymore and *Rb* levels increased in most mice along with decreasing levels of expression of the cell cycle genes *Ccne2*, *Pcna*, and *Mcm3*, as well as genes coding for activating E2Fs and RB family proteins (**Figs 4C** and **S6**). The RB protein was consistently detectable by immunohistochemistry on tumor sections 3 weeks after dox removal (**Fig 4D**), cells in the tumors were larger, mitotic activity less prominent (**Fig 4E**), and necrotic areas increased (**Fig 4F**). Immunostaining for Ki67 showed decreased proliferation at the 3 weeks timepoint (**Fig 4G and 4H**). Overall, while GFP levels decrease rapidly (a few days) in sh*Rb* mice off dox, re-expression of RB in late-stage pituitary tumor in sh*Rb* mice takes 2–3 weeks and is accompanied by decreased proliferation.

## RB re-expression in RB-deficient cancer cells blocks long-term proliferation

The relatively slow re-expression of RB upon dox removal in sh*Rb* mice after 16–19 weeks of dox treatment made it difficult to evaluate the long-term consequences of RB re-expression because at this stage many sh*Rb* mice are dying from their tumors (**Fig 2A**). To circumvent this issue, we chose to study mice with dox treatment for just 6 weeks before dox removal. At this time point, the monotonous growth pattern observed on tissue sections is characteristic of adenomas of the adenohypophysis, with loss of RB and elevated proliferative indices (**S7 Fig**). In this context, RB re-expression can begin just before the first external phenotypic differences can be observed in dox-treated mice, allowing us to age +/-dox mice for many more weeks. We analyzed the sh*Rb* mice 6 and 12 weeks after dox removal (i.e., 12 and 18 weeks after initial RB knockdown, hereafter referred to as 12w-res and 18w-res) (**Fig 5A**). We found that 12w-res mice exhibited phenotypes that were somewhat in between the severity found in mice at 6 weeks +dox and mice at 12 weeks +dox (**Figs 5B, S8A** and **S8B**). At 18 weeks, we observed the expected increase in phenotype severity in the 18w +dox mice, while 18w-res mice showed a continued stabilization of both external and internal phenotypes, but not fully reversed to the extent of control mice (**Figs 5B, S8A** and **S8B**). Cataract was still detectable in the eyes of 18w-res mice (**S8C** and **S8D Fig**). 18w-res mice did not show the weight loss observed in 18w +dox mice (**Figs 5C** and **S9**). Remarkably, we observed no incidences of thyroid tumor growth in these rescue mice (**Fig 5B**). In contrast, we found pituitary tumors in all 12w-res and 18w-res mice. These tumors showed reduced weight compared to 18w +dox treated mice, but still show an increased weight when compared to control pituitaries (**Figs 5B** and **S10A**). Cell size increased in pituitary tumors of 12w-res mice and larger necrotic areas were detected (**S10B–S10E Fig**), both of which may contribute in addition to delayed Rb re-expression upon dox removal to the continued increased pituitary weight. Staining for Ki67 on pituitary sections showed positive signal in the *pars intermedia* and *pars distalis* regions at 6w +dox (**Fig 5D and 5E**). High Ki67 levels were maintained at 12w +dox and 18w +dox but 12w-res and 18w-res tumors showed complete elimination of positive signal for Ki67 (**Fig 5D** and **5E**). The absence of regions where Ki67 is still detected indicates that RB re-introduction occurs throughout the tumor lesions. Thus, RB re-expression in this context results in a long-term cell cycle arrest.

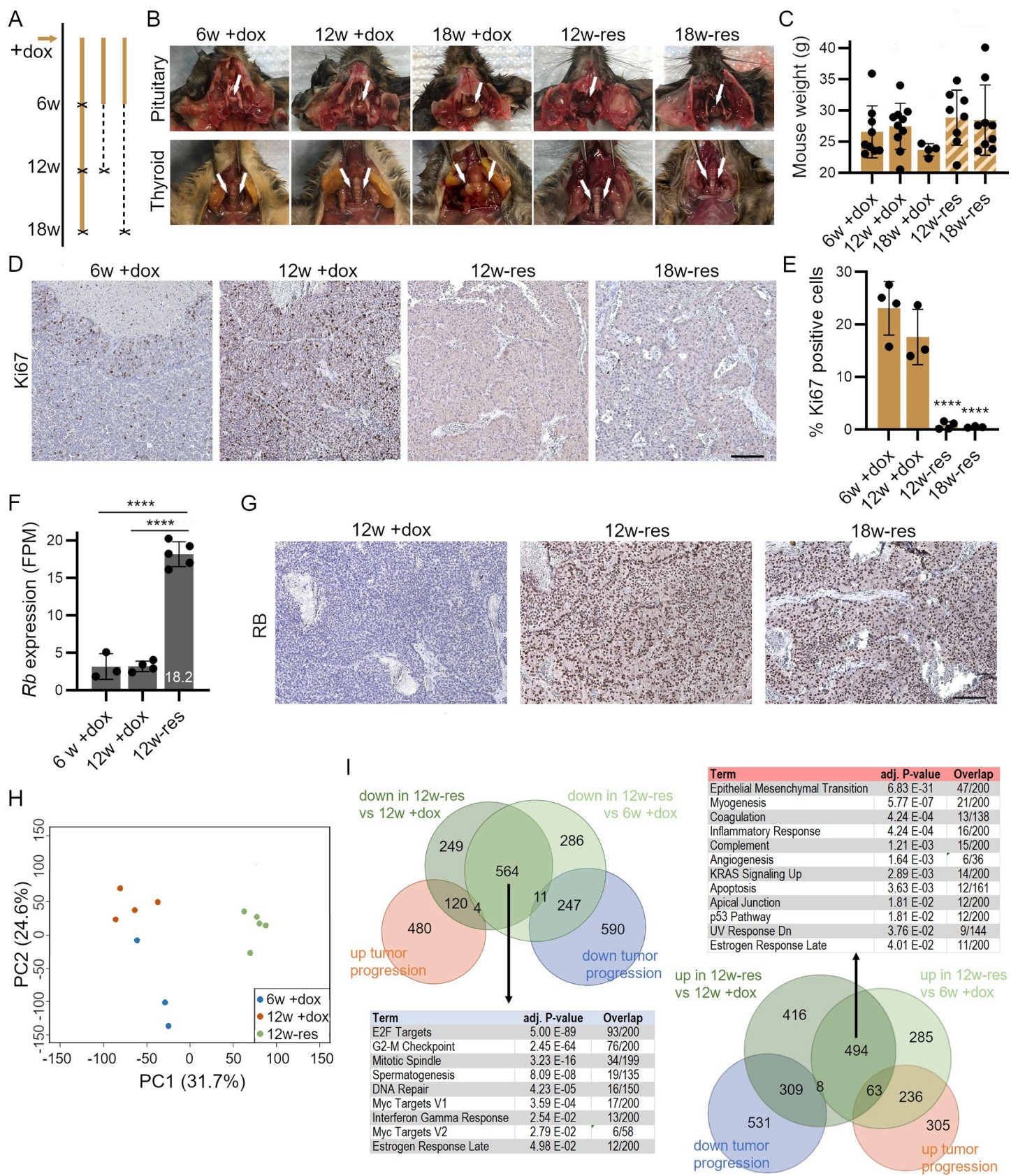

**Fig 5. Sustained cell cycle arrest upon RB re-expression in pituitary adenomas. A.** Timeline diagram of the three different treatment groups of mice (w, weeks). X, time points where mice are euthanized and samples collected. —, periods where doxycycline (dox) is taken away. **B.** Representative images of pituitary and thyroid glands in sh*Rb* mice at different time points of dox treatment (+dox) and dox release (-res). 12w-res, 6w +dox and 6w -dox; 18w-res, 6w +dox and 12w -dox. **C.** Weight of sh*Rb* mice after different time points of dox treatment and dox release (the observed trend is not significant by one-way ANOVA). **D.** Representative images of Ki67 immunostaining on sections from pituitary tumors in sh*Rb* mice at different time points of dox treatment and dox release. Scale bar, 100μm. **E.** Quantification of Ki67 immunostaining as in (D) (n = 2000–3000 cells per section) (one-way ANOVA analysis, significance compared to 6w+dox is indicated; **** = p<0.0001). **F.** Expression of *Rb* in pituitary samples after 6w +dox, 12 w +dox, and 12w-res, determined by RNA-seq analysis (DEseq2; **** = p.adj.<0.0001). **G.** Representative images of RB immunostaining on pituitary tumor sections at different time points of dox treatment and dox release. Scale bar, 100μm. **H.** Principal component analysis (PCA) of RNA-seq. **I.** Venn diagram of differentially-expressed (DE) genes (abs. log2FC>1, p.adj.>0.05) in different conditions and MSigDB Hallmark 2020 enrichment of DE genes independent of tumor progression. Shown are hallmarks with p.adj.<0.05, analyzed with Enrichr.

To profile these pituitary tumors for changes in transcriptional program related to RB re-expression, we performed RNA sequencing on 6w+dox, 12w+dox and 12w-res pituitary tumor samples (**S4 Table**). 604 genes were upregulated and 848 genes downregulated during pituitary tumor progression (6w+dox *vs*. 12w+dox) (**S11A Fig**). This analysis did not detect any increase in cell cycle activity but suggested changes in cell migration, cell adhesion, and differentiation during tumor progression (**S11B and S11C Fig**). When we compared 6w/12w +dox to 12w-res tumors, we observed a 5.7-fold increase in *Rb* expression (**Fig 5F**), which was visible by immunostaining on tumor sections (**Fig 5G**). Comparing the three conditions using principal component analysis (PCA) showed that the re-expression of RB drives the variation along PC1, with genes driving this PC being enriched for cell cycle regulation and repair processes (**Fig 5H** and **S5 Table**). PC2 captures tumor progression between the 6w+dox and 12w+dox time points, with an enrichment for genes involved in metabolic processes, hypoxia response and many signaling pathways, with the rescue group situated in between both groups (**Fig 5H**). This was reflected by overlapping differentially expressed (DE) genes in the three conditions (**Fig 5I**). We also identified DE genes that are independent of tumor progression, and likely represent the direct effect of RB re-expression on the transcriptome (**Fig 5I**). We noted enrichment for terms related to interferon gamma signaling and inflammation in the DE genes. Inhibition of CDK4/6 and RB activation have been linked to immune responses, including interferon signaling ([34,35] and references therein). Immunostaining for CD45 indeed revealed a trend towards increasing infiltration of immune cells into the tumor area upon RB reintroduction (**S12A and S12B Fig**). In the overlapping, tumor progression-independent, DE genes, we also found enrichment of direct RB targets in cell cycle progression in the downregulated fraction, suggesting that RB re-expression is re-establishing RB function as a cell cycle regulator. Genes in the upregulated fraction were enriched in processes related to various pathways, including apoptosis, and the p53 pathway, possibly reflecting tumor restructuring including appearance of necrotic areas upon RB reintroduction. Immunohistochemistry for cleaved caspase 3 (CC3, a marker of apoptosis) showed mostly apoptosis in necrotic areas (**S13A Fig**). P53 expression itself was downregulated upon tumor development and remained unchanged in 12w+dox and 12w-res samples (**S13B Fig**). The p53 target p21 was downregulated upon rescue and other downregulated consensus p53 targets (in total 24) involved genes involved in DNA damage response or the G1/S transition (e.g., *Pcna*, *E2f7*, *Pidd1*, *Rad51c*); upregulated genes (in total 32) were enriched in regulation of differentiation, response to chemical stress, wound healing, and extracellular matrix organization (e.g., *Notch1*, *Hes1*, *Cav1*, *Serpine1*, *Itga3*) (**S13C–S13E Fig**). In conclusion, re-expression of RB at early stages of the development of pituitary adenoma in mice leads to sustained inhibition of proliferation.

## Discussion

Loss of RB function is a frequent event in human cancer. Here we describe the development of a new mouse allele with inducible and reversible RB knock-down. In this model, RB knock-down results in efficient tumor development in the pituitary and thyroid glands, similar to

what has been described in *Rb* knockout mice [24,33,36–39]. We investigated the consequences of endogenous RB restoration in these tumors initiated by loss of RB and found that RB depletion is required for their continuous growth.

With the inducible shRNA approach, it was initially unclear whether the extent of RB knock-down would be sufficient to initiate cancer. This has not been an issue for other tumor suppressors using the same approach [29,40] and it is likely that cells with the greatest RB knock-down undergo selection and expansion *in vivo*, thereby leading to cancer development. It remains unknown, however, whether the 70–80% RB knock-down and the inducible shRNA approach will be sufficient to recapitulate the developmental phenotypes observed in *Rb* knockout embryos and newborn mice [32,33,41,42]. We did not investigate in detail other phenotypes previously described in RB deficient adult mice, such as liver steatosis or adrenal gland tumors [37], but the time frame of death from pituitary tumor development in sh*Rb* mice when dox treatment is started in young adults is strikingly similar to what has been observed upon *Rb* deletion specifically in the pituitary gland at early stages of development [43], indicating that the inducible knock-down approach is highly effective. We note that in our mouse model only one shRNA sequence was used to knock-down RB. Because sh*Rb* mice treated with dox did not display any novel phenotypes compared to *Rb* knockout mice, possible off-target effects of our chosen shRNA molecule may be minimal, but possible off-target effects cannot be excluded and may become more evident in other settings. Overall, our observations indicate that sh*Rb* mice provide a reliable system to investigate RB loss of function *in vivo*. RB has been associated with the regulation of cell plasticity and cell fate decisions in several contexts [18,44–46]. Future experiments with tissue-specific promoters to express rtTA as well as combination of alleles in other genes will help investigate specific mechanisms of action of RB in different contexts in mice, during developmental programs and in various cancer types.

A major advantage of the inducible shRNA platform is the possibility to turn expression of a gene off and on at will with doxycycline. The stability of the targeted protein, doxycycline penetration in tissues and its clearance rates (for example in the brain [47]), and whether cells divide or not may be contributing factors to the efficiency and the kinetics of the knock-down and re-expression. Comparing various inducible shRNA models may help identify key elements to improve this approach in mice. Knock-down of the tumor suppressor APC in the intestine resulted in strong phenotypes that could be rapidly reversed upon removal of dox for 4 days; similarly, APC restoration can drive the regression of intestinal adenoma within a few weeks [48]. The time scale for detectable re-expression of RB in our model was 2–3 weeks, while GFP expression disappeared already after a few days. We speculate that the long-term *in vivo* knockdown leads to a positive feedback loop, inducing repressing factors on the *Rb* locus itself or regulation of factors needed for efficient expression of RB. These circumstances may first need to be overcome before we can observe efficient induction of RB. The kinetics we observed are amenable to cancer studies *in vivo* but may be too slow for studies focusing on more acute phenotypes. In that case, the generation of mice in which RB is fused to specific degrons [49,50] may be needed.

Human pituitary tumors often have alterations in the RB pathway [51–53] and pituitary tumors in *shRb* mice may provide a relevant model for this human cancer. When we investigated the consequences of RB restoration in pituitary tumors initiated by loss of RB, we found that this restoration led to long-term cell cycle arrest. In contrast, in other studies, re-expression of RB in lung tumors driven by oncogenic K-Ras only led to a transient cell cycle arrest because of Cdk2 activation downstream of K-Ras signaling leading to RB hyperphosphorylation [18]. These observations suggest that pituitary cancer cells do not naturally evolve to acquire high CDK2 activity (or other compensatory activity) during tumor progression. These

observations also raise the question of when RB activation will be sufficient to stop the proliferation of cancer cells, including in the context of RB wild-type tumors treated with CDK4/6 inhibitors. It is likely that, similar to K-Ras, activation of oncogenes will activate CDK4/6 and/or CDK2 via several mechanisms, suggesting that a combination of inhibitors targeting these signaling pathways may be required for efficient cell cycle arrest [54,55]. RB may play different tumor suppressor roles in different cancer types, and sh*Rb* mice will be useful in the future to examine in more detail the molecular consequences of RB inactivation and re-activation in cells in multiple contexts *in vivo*.

## Methods

### Ethics statement

Mice were maintained according to practices prescribed by the NIH at Stanford's Research Animal Facility (protocol #31808). Additional accreditation of Stanford animal research facilities was provided by the Association for Assessment and Accreditation of Laboratory Animal Care (AAALAC).

### Cell culture

NIH3T3 and C2C12 cells were grown in DMEM high glucose supplemented with 10% bovine growth serum and penicillin-streptomycin-glutamine. KH2 mouse embryonic stem cells (mESCs) were grown on a feeder layer (irradiated DR4 MEFs) in Knockout DMEM supplemented with 15% fetal bovine serum (Hyclone, SH30070.03E), penicillin-streptomycin-glutamine, 10 μM ß-Mercaptoethanol, and leukemia inhibitory factor (LIF).

For karyotyping, mESCs were cultured in presence of 0.6 μg/mL colcemid (Gibco) for one hour. Cells were then harvested and resuspended in 0.075 M KCl and after 15 minutes they were fixed with a mix of methanol and acetic acid (3:1). Metaphase spreads were prepared by dropping the fixed cells onto glass slides. Glass slides were baked at 90˚C and subsequently stained with Giemsa (Thermo Fisher) after incubation in 0.05% Trypsin and 0.72% NaCl.

### Generation of an inducible sh*Rb* expressing mouse line

Several shRNAs were tested for efficiency as described [27]. The target sequences for the shRNA molecules are shown in (**S1 Table**). Briefly, shRNAs were cloned into pLMP and C2C12 or NIH3T3 cells were tested for RB knockdown efficiency by immunoblot (C2C12) and RT-qPCR (NIH3T3). For targeting KH2 mESCs, shRNAs were cloned into cTGM as described in [27], but with replacement of the miR30 backbone with miR-E backbone as described in [28]. To introduce the inducible shRNA cassette into KH2 mESCs, KH2 cells were transfected with pCAGs-FlpE and cTGM with inserted shRNA sequence using Lipofectamine 2000 (Thermo Fisher) according to suppliers' protocol. Injection of ES cells in blastocysts and embryo transfer were performed by the Transgenic Knockout and Tumor Model Service (TKTC) center at Stanford.

### Animal studies

Doxycycline (dox) treatment (2 mg/mL + 0.7% sucrose, Sigma) was administered through the drinking water supply (Innovive) using brown shaded bottles to avoid degradation by light. Bottles were refilled weekly with a new solution and used bottles were swapped out every 4 weeks to avoid bacterial growth. Sucrose was not added to the water supply of control mice or for RB re-introduction experiments (no dox).

Genotyping was performed by polymerase chain reaction (PCR) and agarose gel (1%) analysis of DNA extracted from tail tissue (Bimake Mouse Direct PCR Kit) for *Col1A1* and *Rosa26* (**S6 Table**).

Mouse weights were measured with a digital scale post euthanasia in a carbon dioxide gas chamber.

## Immunoblotting

Immunoblot analysis of C2C12 cells was performed with the Rb4.1 antibody (Developmental Studies Hybridoma Bank). For immunoblot analysis on liver extracts, liver tissues were harvested and a 2-step collagenase perfusion was performed to isolate hepatocytes (Gibco). Hepatocyte pellets were lysed with RIPA buffer and sonicated to produce total protein extract. Protein concentrations were determined with Pierce BCA Protein Assay Kit (ThermoFisher) using the provided protocol. Samples were then diluted to a final concentration of 40 mg/mL, and 20 μL were loaded into each well of an 8% SDS-PAGE gel with subsequent electrophoresis. The antibodies used were RB (Abcam ab181616, 1:1000) and Actin (Sigma A2066, 1:2000).

## Immunostaining

Pituitary tissues were harvested and fixed in 4% formaldehyde solution for 24 hours and then stored in 70% ethanol. Samples were sent to Stanford University Histology Services Center to be processed for paraffin embedding and sectioning. Initial sections were stained with hematoxylin and eosin (H&E) performed by the Histology Services Center. Subsequent sections were deparaffinized with Histo-Clear (National Diagnostics), serial ethanol dilutions (2x: 100%, 95%, 70%), and distilled water rinses. For Ki-67, PH3, GFP, CD45, and CC3 (cleaved caspase 3) staining, antigen retrieval was performed in a citric acid based unmasking solution (Vector Laboratories H-3300) by heating samples in a microwave for 2 minutes at full power until boiling and then 12 minutes at 30% power. Antigen retrieval for RB staining was carried out by placing samples for 25 minutes in a 1mM EDTA (pH 8.0) buffer bath kept at 100˚C using a vegetable steamer.

The ImmPRESS Excel Amplified Polymer Staining Kit, Anti-Rabbit IgG, Peroxidase (MP-7601) was used following the manufacture's protocol. Washes were performed with 1X phosphate buffer solution (PBS) + 0.1% Tween 20 solution. Samples were incubated with primary antibodies for at least 12 hours overnight at 4˚C. The following primary antibodies were used and diluted in PBS with 2.5% normal horse serum: Ki67 (Invitrogen SP6, 1:750), PH3 (Cell Signaling #9701, 1:770), RB (Abcam ab181616, 1:1500), CC3 (Cell Signaling #9664S, 1:2000), CD45 (Cell Signaling #70257S, 1:200), GFP (Invitrogen #A-11122, 1:200). Samples were incubated for 1–2 minutes with the provided DAB solution to develop optimal detection intensity, then counterstained with hematoxylin for 15 seconds, dehydrated with serial ethanol dilutions and xylene for 5 minutes each, and mounted with glass coverslips.

Images of samples were taken using a digital microscope (Keyence, BZ-X710). Quantification of Ki67 and CD45 immunostaining was performed using QuPath [56]. Region of interests (ROI) were free drawn to include 2000–3000 cells in 20x magnification images and parameters were set to detect OD max of cells with single thresholds. Each sample was then evaluated by eye and confirmed that the detection run was sufficient before accepting the given positive %. Thresholds were changed to multiple as needed for samples where background staining was too dark, and only positive detections of the highest threshold was used in calculations for these specific samples.

Quantification of cell size and necrosis was also performed on QuPath on images of samples. Cell detection of ROIs generated a histogram of cell area and distribution in samples,

which was used to calculate the mean of cell sizes for each section. The wand tool was used to determine the entire area of the tumor, then regions of necrosis were drawn with the wand tool and these areas were totaled up and divided by tumor area to calculate a percentage of necrosis.

## RNA extraction, cDNA synthesis, and RT-qPCR analyses

Liver and pituitary tissues were collected at determined time points of treatment and immediately snap frozen in dry ice and stored at -80˚C until further processing. RNA was extracted using Qiagen RNeasy Plus Mini Kit according to the manufacturer's protocol. 1μg of extracted RNA was used as starting material for cDNA synthesis using ProtoScript First Strand cDNA Synthesis Kit (E6300S) and the provided randomized primer mix and quick protocol. The cDNA product was then diluted 1:20 with water and used for RT-qPCR using Bio-Rad iTaq Universal SYBR Green Supermix on a Bio-Rad CFX384 Real-Time PCR Detection System (for primers see **S6 Table**). Data from runs were exported and calculations computed in Microsoft Excel. Samples were run in triplicates and normalized to *Rps13* expression. Mouse primer sequences used were ordered from Elim Biopharmaceuticals Inc.

## RNA sequencing analysis

Pituitaries were snap-frozen in dry ice and stored at -80˚C. RNA isolation, mRNA library preparation and sequencing on a NovaSeq6000 in a paired-end 150bp run were performed by Novogene Co., Ltd. Obtained reads were mapped to mouse reference genome mm10 with STAR2.5.1b using default settings. Genes from the Refseq database that have at least 10 reads in more than 50% of all samples of one condition, were used for further analysis. Differentially expressed (DE) genes were obtained using DEseq2 (p.adj.<0.05, absolute log2FC>1) [57]. Enriched MSigDB hallmarks were determined using EnrichR [58] and enriched GO terms obtained using GOrilla [59]. REVIGO [60] was used to visualize GO terms. The Gene Expression Omnibus accession number for the RNA-seq data reported in this paper is GSE173749.

## Statistical analyses

Statistical significance was assayed with GraphPad Prism software. The specific tests used are indicated in the figure legends.

## Supporting information

**S1 Fig. Characterization of mouse embryonic stem cell clones with inducible RB knockdown. A.** Immunoblot analysis on protein extracts from mouse C2C12 cells stably expressing different shRNA against *Rb*. Tubulin serves as a loading control. *, non-specific band. **B.** RT-qPCR on RNA isolated from mouse NIH3T3 cells stably expressing different shRNAs against *Rb*. Expression is normalized to a control shRNA against Renilla luciferase (Ren). **C.** RT-qPCR analysis of Rb levels in different mouse embryonic stem cells (mESC) clones with inducible sh*Rb*5 treated with doxycycline (dox) normalized against untreated controls. **D.** Representative images of the different mESC clones with inducible sh*Rb*5 under dox treatment. * marks clones that were euploid by karyotyping. Scale bar, 100μm. Note the induction of GFP as visualized by fluorescence. Clone sh*Rb*5-4 was selected for the generation of mice.
(TIF)

**S2 Fig. GFP and RB expression in different tissues of sh*Rb* mice with or without doxycycline. A.** Mice were analyzed after 10 days of doxycycline (dox) treatment and GFP expression was detected by fluorescence. Representative brightfield (BF) and fluorescence (GFP) images

are shown for two sh*Rb* mice +dox. An sh*Rb* mouse without dox treatment (-dox) was used as a control. White arrows point to the pituitary gland. Scale bar, 4mm. **B.** Mice were analyzed after 21 days +dox and GFP expression was detected by fluorescence. Representative images are shown for liver, lung, brain and tail tissue. Scale bar, 4mm. **C,D.** Representative images of immunohistochemistry analysis for GFP in the retina (C) and RB in the liver liver (D) on sections from mice treated for 21 days +dox compared to no treatment. Scale bar, 100μm.
(TIF)

**S3 Fig. Comparison of survival and body weight upon RB knock-down separated by sex. A and B.** Survival curves of males (A) and females (B) with *Rb* knock-down (doxycycline treatment, dox) and in control mice. Significance was calculated using a Log-rank (Mantel-cox) test. **C and D.** Weight of males (C) and females (D) at different time points (12/13, 15/16, and 18/19 weeks) after *Rb* knock-down and in control untreated mice (one-way ANOVA against control: **, $p<0.01$; ***, $p<0.01$; ****, $p<0.0001$).
(TIF)

**S4 Fig. Representative images of various phenotypes observed in sh*Rb* mice. A.** Images depicting kyphosis, alopecia, and greying of fur on belly of *shRb* mice treated with doxycycline (dox) for 18–19 weeks compared to control untreated mice. **B.** Representative images of eyes in dox treated sh*Rb* mice and control mice and H&E (hematoxylin and eosin) staining of eyes from 18w dox treated sh*Rb* mice and control mice. Note the intact retina and the lens defects. Scale bar, 100μm. **C.** Representative images (H&E staining) of small intestine H&E staining from 18w dox treated sh*Rb* mice. No gross pathological defects were observed. Scale bar, 100μm.
(TIF)

**S5 Fig. Development of thyroid tumors in sh*Rb* mice.** Representative images from H&E (hematoxylin and eosin) stained sections from thyroid glands in control and dox-treated sh*Rb* mice (19 weeks). Scale bar, 100μm.
(TIF)

**S6 Fig. Expression of Rb pathway related genes upon RB reintroduction.** RT-qPCR analysis 2 weeks after RB reintroduction normalized to *Rps13*. *Rbl1* and *Rbl2* code for the RB family members p107 and p130, respectively. *E2f1/2/3* code for activating E2Fs. Data were analyzed using a two-tailed students t-test; *, $p<0.05$; **, $p<0.01$; ***, $p<0.001$.
(TIF)

**S7 Fig. RB expression and proliferation in pituitary glands of sh*Rb* mice after 6 weeks on doxycycline.** Immunostaining for RB, Ki67, and phospho-histone 3 (PH3) on sections from the pituitary glands of a control mouse (n = 1) and an sh*Rb* mouse after 6 weeks of doxycycline (dox) treatment (representative of n = 4 mice). Scale bars, 50μm (RB) and 100μm (Ki67 & PH3).
(TIF)

**S8 Fig. Phenotypes of sh*Rb* mice upon RB re-expression. A.** Representative images of sh*Rb* mice after 6 weeks of doxycycline (6w +dox) treatment and then continued dox treatment or dox removal. **B.** Representative images of the internal abdominal cavity in mice as in (A). **C.** Representative images of eyes in different dox treatment groups. **D.** Representative images of IHC H&E sections of eyes in different dox treatment groups. Scale bar, 400μm.
(TIF)

**S9 Fig. Comparison of weights of sh*Rb* mice in early- and long-term RB reintroduction cohorts separated by sex. A and B.** Weight of males (A) and females (B) at different time points after *Rb* knock-down (with doxycycline, dox) or re-expression (w, weeks) (one-way ANOVA: *, p<0.05). nd, not determined (no female mice at this time point).
(TIF)

**S10 Fig. Characterization of tumor size, cell size, and necrosis in pituitary tumors after long-term sustained RB re-expression. A.** Weights of 18w-res pituitary tumors compared to control pituitary glands and 18/19w RB KD tumors (one-way ANOVA analysis: *, p<0.05; **, p<0.01; ****, p<0.0001). **B.** Representative cell area histograms obtained from Qupath cell detection analysis in computing cell size of 12w+dox (12 weeks on doxycycline) and 12w-res (6 weeks on doxycycline and 6 weeks off) tumor sections. **C.** Cell size averages (arbitrary units) of tumor sections obtained from histogram analysis in (B) (unpaired t test; *, p<0.05; **, p<0.01). **D.** Representative images of Hematoxylin-stained sections used to mark areas of necrosis in tumors using the Qupath wand tool. Scale bar, 100μm. **E.** Computed % area of necrosis in tumor samples from (D) in 12w+dox and 12-res sections (unpaired t test).
(TIF)

**S11 Fig. RNA sequencing of pituitary lesions in sh*Rb* mice with 6 weeks and 12 weeks dox treatment. A.** Heatmap of differentially-expressed (DE) genes by RNA-seq analysis (DEseq2: log2FC >1; p.adj.<0.05) in pituitary lesions from sh*Rb* mice after 6 weeks or 12 weeks in doxy-cycline (dox) treatment. **B.** MSigDB Hallmark 2020 enrichment of downregulated genes during tumor progression from the RNA-seq as in (A). Shown are Hallmarks with p.adj.<0.05, analyzed with Enrichr (Kuleshov *et al.* [58]). Upregulated genes are not enriched for any of the Hallmark datasets. **C.** Summary of enriched GO Processes in downregulated genes during tumor progression from the RNA-seq as in (A), using Revigo (Supek *et al.* [60]). Upregulated genes were only enriched for transmembrane transport.
(TIF)

**S12 Fig. CD45 staining in pituitary tumors upon long-term RB reintroduction. A.** Representative images of CD45 staining on sections from different treatment groups. Scale bar, 100μm. **B.** Quantification of % positive CD45 cells on samples of groups obtained in (A) (unpaired t-test was performed, no significance noted).
(TIF)

**S13 Fig. Expression of p53 target genes upon long-term RB re-introduction. A.** Immunostaining for the apoptosis marker cleaved caspase 3 (CC3) on two 12w-res tumor sections (counterstained with hematoxylin). Scale bar, 100μm. **B,C.** *Trp53* (B) and *Cdkn1a* (C) expression levels from RNA-seq data (****; DEseq2: padj<0.0001, log2FC>|1|). **D.** Heatmap of RNA-seq results from differentially expressed consensus p53 targets as defined by Fischer et al. (2017) [61]. **E.** List of differentially-expressed genes from Heatmap in (D).
(TIF)

**S1 Table. shRNA target sites for RB knock-down in mouse cells.**
(XLSX)

**S2 Table. Phenotypes observed in sh*Rb* mice at different time points of constant dox treatment**
(XLSX)

**S3 Table. Comparison of phenotypes in *Rb* knockout mice and sh*Rb* mice.**
(XLSX)

**S4 Table. RNA sequencing of pituitaries from 6-weeks dox-treated sh*Rb* mice (E16,17, 18), 12-weeks dox-treated sh*Rb* mice (E1,E2, E4, E5) and 6-weeks dox plus 6-weeks no dox sh*Rb* mice (E8,E9, E11, E12, E13)**
(XLSX)

**S5 Table. GO Terms of Biological Processes of Top 100 genes driving PC1 or PC2.**
(XLSX)

**S6 Table. Primers for genotyping and RT-qPCR**
(XLSX)

**S1 Source Data. All raw data for all panels with graphs are compiled into one document.**
(XLSX)

## Acknowledgments

We thank Pauline Chu from the histology facility for her help with tissue sections, Hong Zeng from the Transgenic Knockout and Tumor Model Service and all the members of the Sage lab for their help and support throughout this study, especially Thuyen Nguyen.

## Author Contributions

**Conceptualization:** Alex Doan, Julia Arand, Julien Sage.

**Formal analysis:** Alex Doan, Julia Arand, Diana Gong, Alexandros P. Drainas, Yan Ting Shue, Myung Chang Lee, Hannes Vogel.

**Funding acquisition:** Julien Sage.

**Investigation:** Alex Doan, Julia Arand, Shuyuan Zhang, David M. Walter.

**Project administration:** Julien Sage.

**Resources:** Lukas E. Dow.

**Supervision:** Julia Arand, David M. Feldser, Jan M. Skotheim, Julien Sage.

**Validation:** Andrea C. Chaikovsky.

**Visualization:** Alex Doan, Julia Arand, Myung Chang Lee, Julien Sage.

**Writing – original draft:** Alex Doan, Julia Arand, Julien Sage.

**Writing – review & editing:** Diana Gong, Alexandros P. Drainas, Yan Ting Shue, Myung Chang Lee, Shuyuan Zhang, David M. Walter, Andrea C. Chaikovsky, David M. Feldser, Hannes Vogel, Lukas E. Dow, Jan M. Skotheim, Julien Sage.

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
