## [Editor Report · Decision Letter 0]

7 Aug 2021

Dear Dr Sage,

Thank you very much for submitting your Research Article entitled 'RB depletion is required for the continuous growth of tumors initiated by loss of RB' to PLOS Genetics.

The manuscript was fully evaluated at the editorial level and by independent peer reviewers assigned by Review Commons.  The reviewers appreciated the attention to an important problem, but raised some substantial concerns about the current manuscript. Based on the reviews, we will not be able to accept this version of the manuscript, but we would be willing to review a revised version of the paper as you propose in your response letter.. We cannot, of course, promise publication at that time.

Should you decide to revise the manuscript for further consideration here, your revisions should address the specific points made by each reviewer as you propose. We will also require a detailed list of your responses to the review comments and a description of the changes you have made in the manuscript.

If you decide to revise the manuscript for further consideration at PLOS Genetics, please aim to resubmit within the next 60 days, unless it will take extra time to address the concerns of the reviewers, in which case we would appreciate an expected resubmission date by email to plosgenetics@plos.org.

[LINK]

We are sorry that we cannot be more positive about your manuscript at this stage. Please do not hesitate to contact us if you have any concerns or questions.

Yours sincerely,

Bruce E. Clurman

Associate Editor

PLOS Genetics

David Kwiatkowski

Section Editor: Cancer Genetics

PLOS Genetics

We have carefully read the reviews obtained via Review Commons as well as your response. You manuscript is certainly appropriate for PloS Genetics. We thus ask you to proceed with a revision along the lines that you propose in your response to the reviews and we would be happy to consider the revised paper.

---

## [Decision Letter · Decision Letter 1]

11 Nov 2021

Dear Dr Sage,

We are pleased to inform you that your manuscript entitled "RB depletion is required for the continuous growth of tumors initiated by loss of RB" has been editorially accepted for publication in PLOS Genetics. Congratulations!

Yours sincerely,

Bruce E. Clurman

Associate Editor

PLOS Genetics

David Kwiatkowski

Section Editor: Cancer Genetics

PLOS Genetics

Comments from the reviewers (if applicable):

Reviewer's Responses to Questions

**Comments to the Authors:**

Reviewer #1: The manuscript by Alex Doan, Julia Arand et al., describes a novel DOX-inducible shRNA system

to silence and re-express the tumor suppressor Rb; the analysis is exceptionally well performed, the results of great interest with clear potential for follow-up/new discoveries using this model.

The Authors addressed my concerns and I strongly recommend this important manuscript for publication.

Reviewer #2: .

**Have all data underlying the figures and results presented in the manuscript been provided?**

Reviewer #1: Yes

Reviewer #2: Yes

PLOS authors have the option to publish the peer review history of their article (what does this mean?). If published, this will include your full peer review and any attached files.

Reviewer #1: No

Reviewer #2: No

**Data Deposition**

http://datadryad.org/submit?journalID=pgenetics&manu=PGENETICS-D-21-00958R1

**Press Queries**

---

## [Editor Report · Acceptance letter]

25 Nov 2021

PGENETICS-D-21-00958R1 

RB depletion is required for the continuous growth of tumors initiated by loss of RB 

Dear Dr Sage, 

We are pleased to inform you that your manuscript entitled "RB depletion is required for the continuous growth of tumors initiated by loss of RB" has been formally accepted for publication in PLOS Genetics! Your manuscript is now with our production department and you will be notified of the publication date in due course.

With kind regards,

Agnes Pap

PLOS Genetics

On behalf of:
